# Association between INDELs in MicroRNAs and Susceptibility to Gastric Cancer in Amazonian Population

**DOI:** 10.3390/genes14010060

**Published:** 2022-12-24

**Authors:** Antonio A. C. Modesto, Milene R. de Moraes, Cristina M. D. Valente, Marta S. C. R. Costa, Diana F. da V. B. Leal, Esdras E. B. Pereira, Marianne R. Fernandes, Jhully A. dos S. Pinheiro, Karla B. C. C. Pantoja, Fabiano C. Moreira, Rommel M. R. Burbano, Paulo P. de Assumpção, Ney P. C. dos Santos, Sidney E. B. dos Santos

**Affiliations:** 1Núcleo de Pesquisas em Oncologia, Universidade Federal do Pará, R. dos Mundurucus 4487, Guamá, Belém 66073-000, Brazil; 2Laboratório de Genética Humana e Médica, Instituto de Ciências Biológicas, Universidade Federal do Pará, Belém 66073-000, Brazil; 3Hospital Ophir Loyola, Belém 66063-240, Brazil

**Keywords:** gastric cancer, INDEL, potential biomarker, miRNA, susceptibility

## Abstract

Gastric cancer (GC) is a multifactorial, complex, and aggressive disease with a prevalence of one million new cases and high global mortality. Factors such as genetic, epigenetic, and environmental changes contribute to the onset and progression of the disease. Identification of INDELs in miRNA and its target sites in current studies showed an important role in the development of cancer. In GC, miRNAs act as oncogenes or tumor suppressors, favoring important cancer pathways, such as cell proliferation and migration. This work aims to investigate INDELs in the coding region of miRNAs (hsa-miR-302c, hsa-miR-548AJ-2, hsa-miR-4274, hsa-miR-630, hsa-miR-516B-2, hsa-miR-4463, hsa-miR-3945, hsa-miR-548H_4, hsa-miR-920, has-mir-3171, and hsa-miR-3652) that may be associated with susceptibility and clinical variants of gastric cancer. For this study, 301 patients with GC and 145 individuals from the control group were selected from an admixed population in the Brazilian Amazon. The results showed the hsa-miR-4463, hsa-miR-3945, hsa-miR-548H_4, hsa-miR-920 and hsa-miR-3652 variants were associated with gastric cancer susceptibility. The hsa-miR-4463 was significantly associated with clinical features of GC such as diffuse gastric tumor histological type, “non-cardia” localization region, and early onset. Our findings indicated that INDELs could be potentially functional genetic variants for gastric cancer risk.

## 1. Introduction

Cancer is a global health problem, with 18 million new cases and 9.6 million deaths per year. It represents about 12% of all causes of death and is the fourth leading cause of death of those aged 70 years or over in most countries [1,2]. Among all types of cancer, gastric cancer (GC) is the fifth most frequently diagnosed cancer and the third most deadly neoplasm worldwide. GC is a multifactorial, complex, and aggressive disease with a number of genetic, epigenetic, and environmental factors [3].

In 2020, over one million new cases of gastric cancer and 769,000 deaths were estimated worldwide. Infection with *Helicobacter pylori* is the primary identified cause of gastric cancer. Other risk factors for gastric cancer include a high intake of salty and smoked food, obesity, alcoholism, and smoking. However, a diet rich in fruits, vegetables, cereals, and seafood acts as a protective factor against GC [4,5]. Recent evidence shows that genetic variants are also an important factor for the tumorigenic process [6,7,8].

Somatic mutations such as INDELs may provide a selective advantage for cell growth and may initiate cancer development, including GC [9]. These genomic alterations in miRNA genes (pri-miRNAs, pre-miRNAs, mature region and SEED region) or target sites (RNAm) lead to post transcriptional regulation of mRNAs and the biogenesis of mature miRNAs and may be related to the risk of diseases. An experimental study of INDELs in miRNAs and their target sites showed the functional role of these polymorphisms in the development of diseases such as GC [10].

Investigating genetic mutation caused by INDELs in a DNA sequence is relevant because they can alter DNA sequences, which potentially results in a structurally and functionally modified protein [5,6,7]. The objective of this work was to investigate INDELs in the coding region of miRNAs that may be associated with susceptibility and clinical variants of gastric cancer.

## 2. Material and Methods

### 2.1. Ethical Declaration

The study protocol was approved by the ethics committee of Núcleo de Pesquisas em Oncologia da Universidade Federal do Pará (CAE: 30352920.0.0000.5634). Written informed consent was obtained from all study subjects.

### 2.2. Sample Selection

This is a case-control study which included 446 participants that were randomly selected from a public healthcare center in the Brazilian Amazon. In total, 301 patients who were diagnosed with gastric cancer in Hospital João Barros Barreto were included in the case group and 145 participants were admitted to the same hospital without any oncological disease and were included in control group. The analyses were performed from the collection of 5 mL of whole blood from each patient for the next step of DNA extraction.

### 2.3. DNA Extraction and Quantification

The DNA was extracted using the phenol–chloroform method and stored at −24 °C [11]. DNA quantity and quality were measured with a NanoDrop ND-1000 spectrophotometer (NanoDrop Technologies, Wilmington, DE, USA).

### 2.4. Selection Polymorphism—INDEL

Eleven INDELs in miRNA genes were selected for their function in several diseases. These miRNAs were found differently expressed in cancer in the literature [12]. INDELs were selected with medium and highest allele frequencies (MAF) ≥ 0.02 MAF based on the 1000 Genomes Project [12] (Table 1).

### 2.5. Genotyping

Multiplex PCR allowed the simultaneous analysis of 11 INDELs. Each primer was labeled (FAM, VIC, or HEX). Amplification of fluorescent PCR was performed in the ABI Verity thermocycler (Life Technologies, Foster City, CA, USA). A single multiplex reaction used the Master Mix QIAGEN Multiplex PCR Kit (Qiagen, Hilden, Germany) according to the manufacturer’s instructions.

For fragment analysis, capillary electrophoresis was carried out, using the ABI 3500 Genetic Analyzer instrument (Thermofisher Scientific, Waltham, MA, USA). For capillary electrophoresis, 1.0 μL of the PCR product was added to 8.5 μL of HI-DI deionized formamide (Life Technologies, Carlsbad, CA, USA) and 0.5 μL of GeneScan 500 LIZ size standard (Life Technologies). The samples were analyzed using both GeneMapper^®^ ID v.3.7 software (Thermofisher Scientific).

### 2.6. Statistical Analysis

The chi-square test was performed to determine whether genotype distributions were within Hardy–Weinberg equilibrium. Logistic regression analyses with Bonferroni correction factor were applied to improve the accuracy of genotypic models and the variables of clinical characteristics. Additional statistical analyses were made using the software RStudio [13]. A *p*-value of <0.05 was considered statistically significant.

## 3. Results

### 3.1. Demographic Characteristics

We analyzed 301 samples from individuals with GC and 145 samples from cancer-free individuals and we compared epidemiological characteristics of gender and age between groups, but no significant differences were found (chi-square test). Table 2 describes the clinical and epidemiological characteristics of the investigated patients. Statistical comparison between case-control characteristics related to gender and age were not significant. Regarding the location of the tumor, the most common type among the samples was the antrum and the least common was the pylorus, and when we classified samples according to Lauren, the diffuse type was the most found. In terms of staging, stage two was the most identified among the samples.

### 3.2. Distribution of Genotypes Associated with Susceptibility to GC

Hardy–Weinberg equilibrium (HWE) analysis was performed on the samples studied to control the variables that could be influenced by genotyping errors. Four variants *MIRNA302C*, *MIRNA630*, *MIRNA3171*, and *MIRNA548AJ_2* were in Hardy–Weinberg disequilibrium (*p* < 0.05) and were removed from the analysis.

The genotypic and allelic frequencies of the groups are described in Table 3. Significant differences were found in variants *miRNA4463*, *miRNA920,* and *miRNA 3652* between cases and controls. The complete table can be found in the Appendix A.

When analyzing the seven INDELs variants regarding the association for susceptibility to GC, three variants were found associated with increased risk for GC (the Del allele of *miRNA4463*, *miRNA3945,* and *miRNA548H_4*) and two variants associated with decreased risk for CG (the Del allele of *miRNA3652* and *miRNA920*), as shown in Table 4. The complete table on susceptibility to gastric cancer can be found in the Appendix A.

### 3.3. Distribution of Genotypes Associated with Clinical Variants in GC

In addition to susceptibility, the clinical variants of the investigated patients were also evaluated, and when evaluating the genetic impact of the seven INDELs variants on the clinical variants categorized into Lauren subtype (diffuse or intestinal), location (cardia and non-cardia), and age (less than 40 years and more than 40 years), a significant association was found only in the *miRNA4463* variant, as shown in Table 5.

There was no correlation of the other variants with the clinical characteristics as no variant had a significant association with tumor staging. The complete table of associations with the clinical variants can be found in the Appendix A.

Regarding the type of gastric adenocarcinoma, significant associations of the Del allele of the *MIRNA4463* gene with an increased risk for the incidence of diffuse gastric adenocarcinoma were observed. By categorizing the tumors in relation to location (cardia and non-cardia), the analyses showed a significant association of the Del allele of the *MIRNA4463* gene with an increased risk of developing the tumor in the “non-cardia” region. When evaluating the allelic distribution in relation to age (less or more than 40 years), it was observed that the Del allele of *MIRNA4463* was associated with a higher incidence in younger patients.

## 4. Discussion

The study of variants in miRNAs is of great importance to understand the pathways that affect cancer since several genomic alterations occur in specific regions of the gene that can modify the functionality and biogenesis of a molecule to be encoded as a microRNA. Additionally, evaluating variants in genes of patients with GC in the northern region of Brazil, a region that presented a high incidence of new cases compared to the general rates elsewhere, is very relevant and may provide valuable data for the elucidation of gastric carcinogenesis [1,9,10,14].

In this case-control study, we analyzed a panel of 11 INDELs in 446 samples, including 301 patients with GC and 145 individuals in the control group, all of them from an admixed population of the Brazilian Amazon. A limiting factor in the study for the control group was the absence of negative upper digestive endoscopy exams in the patients. It is important to highlight that no previous studies associating these 11 INDELs variants with GC were found. Our data demonstrated a significant association regarding the susceptibility between five variants (hsa-miR-3945_rs145931056, hsa-miR-548H_4_rs150141473, hsa-miR-920_rs66686007, hsa-miR-3652_rs62747560, and hsa-miR-4463_rs5877455) and a significant association regarding the clinicopathological data (and hsa-miR-4463_rs5877455).

*MIRNA3945*, located on chromosome 4, presented an INDEL of 12 base pairs in the mature region of the gene. As for the susceptibility analysis, we found a significant association of the altered Del allele of hsa-miR-3945 with an increased risk for the development of GC. Although this miRNA is not described in the literature associated with GC, the miRDB database (database for functional targets of miRNA) points to hsa-miR-3945 as a target of proteins of the Rab family, which is considered a potential indicator of metastasis and prognosis for lung carcinoma [15]; in addition, hsa-miR-3945 is also a target of the TNF receptor protein family that acts in the process of cell proliferation and tumorigenesis [16,17].

The *miR548h-4* variant, located on chromosome 8, showed an INDEL of 5 base pairs in the pre-miRNA region. Regarding susceptibility, we found a significant association of the altered Del allele with an increased risk of GC development (Table 4). This INDEL has been mentioned in several important studies, such as the study with osteosarcoma, in which the analysis negatively correlated the expression of miRNA548 with the expression of *KRAS*, a gene active in the process of tumorigenesis and metastasis of several types of cancer, including gastric adenocarcinoma [18,19,20].

The INDEL *miRNA548H_4* enables changes that influence the functioning of miRNA548, as its underexpression acts by reversing the effects of KRAS silencing, stimulating metastatic growth and migration [18]. Therefore, this study reinforces the hypothesis that the altered Del allele may act on the loss of *miRNA548* function in patients with GC and consequently reverse the effects of KRAS silencing, which stimulates cell growth and the metastatic process of cancer.

Regarding *MIRNA3652*, our data also demonstrate a significant association of the altered Del allele of *miRNA3652* with decreased risk for the development of GC (Table 4). Changes in this gene act on *MIRNA3652,* which plays an important role in regulating the expression of some members of the *BCL2* family. Bcl-2 is an anti-apoptotic protein capable of regulating mitochondrial physiology and cell death. When dysregulated, Bcl2 also impairs sensitivity to apoptosis-inducing anticancer drugs [21]. Therefore, a possible interpretation of the data from this INDEL is that the altered Del allele in *MIRNA3652* is downregulating genes such as *BCL2*, thus contributing to a reduced risk of GC, which corroborates our data.

With an INDEL of 5 base pairs in the pre-miRNA region, *MIRNA920*, located on chromosome 12, showed a significant difference in the susceptibility of cases and controls, in which the altered Del allele of this miRNA showed a decreased risk for GC (Table 4). Research involving this miRNA is associated with hepatocellular carcinoma [22], in which the study associated levels of hsa-miR-920 to the β protein TrCP, that regulates the NFKβ signaling pathway, both involved in the progression of hepatocarcinoma. Moreover, *miR920* acts on several cancer pathways, involving genes of great potential in oncology, such as *CEBPB*, *MYC* and *TGFBR2*, which are related to the proliferation, anti-apoptosis, invasion, and metastasis pathways in pathologies such as colorectal cancer, leukemia, lymphoma, sarcoma, prostate cancer, and neuroblastoma [23]. Therefore, we suggest that the altered Del allele is preventing the expression of *miRNA920* and thus favoring the reduced risk to the GC. Despite the importance that the marker presented in several cancer-associated pathways, further studies of this marker associated with gastric cancer are necessary.

In addition to these variants, our study showed a significant association of the *MIRNA4463* gene with susceptibility to developing GC (Table 4). Regarding the clinical characteristics, our data also showed that the Del allele of *MIRNA4463* had an increased risk for the development of early gastric cancer, and when we categorized the tumors by region of the stomach, we observed that the Del allele of this miRNA presented an increased risk of developing tumors in the non-cardia region. Moreover, when analyzed according to the classification from Lauren, the Del allele of the *MIRNA4463* gene showed a significant difference and was associated with a greater chance of developing diffuse gastric tumors (Table 5).

According to Assumpção (2020), diffuse GC has a high frequency in younger patients and a stronger relevance in the association with genetic than environmental factors, and thus finding a significant association of the Del allele of *MIRNA4463* to diffuse type tumors and associating it with a significant chance of developing GC earlier reinforces the potential of the *miRNA4463* marker since tumors of this type are those that are not associated with precancerous lesions, occur in younger patients, and have a worse prognosis and an invasive growth pattern [24].

The *miRNA4463* is associated with proliferation, migration and inhibition of apoptosis in colorectal cancer (CRC), as it is overexpressed in these types of tissues, with an even greater expression in metastatic RCC tissues. *miRNA4463* targets the tumor suppressor PPP1R12B, which acts by impairing tumorigenesis and metastatic process in CCR [25,26]. In this case, our data suggest that the deletion of this marker is silencing of PPP1R12B tumor suppressor gene, so that would favor oncogenic pathways and, with that, the progression of GC. Therefore, it is understood that this variant may be an excellent candidate as a risk marker for early gastric cancer and with a worse prognosis. In addition, the literature also points out that miRNA4463 influences some types of cancer since its overexpression was associated with shorter overall survival in patients with hepatocellular carcinoma [27] and with clinically relevant subgroups of lymphoma [28]; it is, in addition, a possible diagnostic biomarker for colon cancer, as high levels of expression of this miRNA have been identified in colon tumors [29].

## 5. Conclusions

This study contributed to an increased knowledge on INDELs in microRNA genes in regard to GC development. Further studies in the same population with a greater sample number and in different populations for comparison are needed as well as functional studies focused on other INDELs to better understand the involvement of microRNA gene in gastric cancer. Our findings demonstrate that three INDELs variants in genes encoding microRNAs (hsa-miR-4463, hsa-miR-3945, and hsa-miR-548H_4) are associated with an increased risk of developing GC, whereas the hsa-miR-920 and hsa-miR-3652 variants are associated with a decreased risk of developing GC. These results corroborate the importance of searching for new biomarkers in the genetic and clinical analysis of patients with gastric cancer in the Amazon region.

## Figures and Tables

**Table 1 genes-14-00060-t001:** Technical characteristics of the markers selected in this study.

Gene	ID	Region	Alleles	MAF	Primers	Amplicon
*miRNA302C*	rs199971565	seed	CACTT/C	0.08	F5′GCTTCCAGTTCCATCCATGT3′	253–257 bp
					R5′CTCAGCGTGGTAGTGTTGGA3′	
*miRNA3945*	rs145931056	mature	CCTATGCCCTCC/-	0.28	F5′AGGAGTATCCCCTCGTGGAC3′	145–157 bp
					R5′CAAGAGTCAGGCAAAAACAGG3′	
*miRNA548AJ-2*	rs145326096	mature	AAGT/-	0.39	F5′CTCTTCAATGCTTCCTTGAGGT3′	207–211 bp
					R5′CTGCATGCCAGGAGCTAAGTAT3′	
*miRNA4274*	rs202195689	mature	-/CCC/CCCCA	0.26	F5′TTTTTGTCCTCCAAGCTTCC3′	132–135 bp
					R5′GAACAAGAGAGAGGGCAGGA3′	
*miRNA630*	rs139334001	pre-miRNA	-/TTG	0.42	F5′GGTGACCCCAGAATTGACCCT3′	96–99 bp
					R5′GCCCTCAGGACGCACCTCTG3′	
*miRNA516B-2*	rs10670323	pre-miRNA	-/AAAGA	0.32	F5′CATGCACAGCTATCCAGGAG3′	162–167 bp
					R5′TTGTTCCTGTCCGATAGATGC3′	
*miRNA 4463*	rs5877455	pre-miRNA	-/AG	0.47	F5′TGCCCCTACTTAGCAGTCTCA3′	191–193 bp
					R5′GAGAGGTGGAGAACTGGGATT3′	
*miRNA 920*	rs66686007	pre-miRNA	GTTGT/A	0.12	F5′GCATCAGGACGCTGAACATA3′	215–220 bp
					R5′AATGCAACTTGCTCCAGAGG3′	
*miRNA 3171*	rs35170395	pre-miRNA	-/TA	0.21	F5′CTGTGTGTCTGAGGGGTGAA3′	331–333 bp
					R5′ATCCTGCCACTTTCTGATGG3′	
*miRNA 548H-4*	rs150141473	pre-miRNA	TAAAG/-	0.28	F5′GGAATGGAAAATAGACAAGAAGTGA3′	197–202 bp
					R5′TGGCAAGTGTACCACAGAAAAC3′	
*miRNA 3652*	rs62747560	pre-miRNA	GGGGTGG/-	0.36	F5′ATTGGTGGGTTCATGTTTCC3′	236–246 bp
					R5′CAGAATCACTCACCGAAGGTC3′	

**Table 2 genes-14-00060-t002:** Demographic data of gastric cancer cases and control.

Characteristic	Cases *n* = 301	Controls *n* = 145	*p*-Value
**Age (yr) Median**	49 (18–88)	41.5 (18–65)	0.198
≤40	146 (48.6)	80 (55.1)	
>40	155 (51.4)	65 (44.82)	
**Gender**			0.606
Male	175 (58.4)	81 (55.8)	
Female	126 (41.55)	64 (44.1)	
**Tumor Location**			
Body	60 (20.7)	NA	
Cardia	88 (29.7)	NA	
Antrum	91 (30.7)	NA	
Fundus	31 (10.4)	NA	
Pylorus	6 (2)	NA	
Missing	25 (6.5)	NA	
**Lauren**			
Intestinal	111 (37.5)	NA	
Diffuse	160 (54)	NA	
Missing	30 (8.44)	NA	
**Tumor Staging**			
Stage 1	32 (10.8)	NA	
Stage 2	65 (21.9)	NA	
Stage 3	56 (18.9)	NA	
Stage 4	62 (20.9)	NA	
Missing	86 (27.5)	NA	

**Table 3 genes-14-00060-t003:** Genotypic and allelic frequency of selected INDELs from NCBI database.

Gene	Cases *n* (%)	Controls *n* (%)	*p*-Value	OR (95% CI)
*miRNA516B_2* (rs10670323)				
Ins/Ins	3 (1.0%)	3 (2.1%)		
Ins/Del	57 (19%)	28 (19.4%)	0.66	0.47 (0.09–2.37)
Del/Del	240 (80%)	113 (78.5%)		
Ins	0.10	0.11		
Del	0.89	0.88		
*miRNA4463* (rs5877455)				
Ins/Ins	85 (28.6%)	57 (41.6%)		
Ins/Del	148 (49.8%)	59 (43.1%)	0.02	2.04 (1.13–3.71)
Del/Del	64 (21.5%)	21 (15.3%)		
Ins	0.53	0.63		
Del	0.46	0.36		
*miRNA3945* (rs145931056)				
Ins/Ins	236 (80.3%)	127 (88.8%)		
Ins/Del	54 (18.4%)	14 (9.8%)	0.05	1.08 (0.19–5.96)
Del/Del	4 (1.4%)	2 (1.4%)		
Ins	0.89	0.93		
Del	0.10	0.06		
*miRNA548H_4* (rs150141473)				
Ins/Ins	220 (74.8%)	120 (83.9%)		
Ins/Del	71 (24.1%)	22 (15.4%)	0.09	1.64 (0.17–15.90)
Del/Del	3 (1.0%)	1 (0.7%)		
Ins	0.86	0.91		
Del	0.13	0.08		
*miRNA4274* (rs202195689)				
Ins/Ins	269 (90.9%)	121 (85.2%)		
Ins/Del	25 (8.4%)	21 (14.8%)	0.09	0.54 (0.29–0.99)
Del/Del	2 (0.7%)	0 (0%)		
Ins	0.95	0.92		
Del	0.04	0.07		
*miRNA920* (rs66686007)				
Ins/Ins	269 (91.5%)	119 (82.1%)		
Ins/Del	24 (8.2%)	24 (16.6%)	0.01	1.00 (0.39–1.47)
Del/Del	1 (0.3%)	2 (1.4%)		
Ins	0.95	0.90		
Del	0.04	0.09		
*miRNA3652* (rs62747560)				
Ins/Ins	183 (62.2%)	89 (62.7%)		
Ins/Del	103 (35%)	39 (27.5%)	0.00	1.28 (0.82–2.01)
Del/Del	8 (2.7%)	14 (9.9%)		
Ins	0.79	0.73		
Del	0.20	0.23		

**Table 4 genes-14-00060-t004:** Allelic variants in MIRNAs with significant associations for GC predisposition. Logistic regression analysis with Bonferroni correction.

Gene	Model	OR (95% CI)	*p*-Value
*miRNA4463_rs5877455_*	Ins/Ins vs. Del/Ins + Del/Del	1.78 (1.16–2.71)	0.007
*miRNA3945_rs145931056*	Ins/Ins vs. Del/Ins + Del/Del	1.95 (1.08–3.53)	0.021
*miRNA548H_4_rs150141473*	Ins/Ins vs. Del/Ins + Del/Del	1.75 (1.05–2.95)	0.028
*miRNA920_rs66686007*	Ins/Ins vs. Del/Ins + Del/Del	0,43 (0.24–0.77)	0.004
*miRNA3652_rs62747560*	Ins/Ins + Ins/Del vs. Del/Del	0.26 (0.10–0.62)	0.002

Genotypes Del/Del = homozygous deletion, Del/Ins = heterozygous, and Ins/Ins = homozygous insertion.

**Table 5 genes-14-00060-t005:** Significant associations between INDEL *miRNA4463* and clinical variants in GC.

Gene	Model	Categorization	OR (95% CI)	*p*-Value
*miRNA4463_rs5877455_*	Ins/Ins vs. Del/Ins + Del/Del	Diffuse or Intestinal type	2.30 (1.27–4.18)	0.004
*miRNA4463_rs5877455_*	Ins/Ins vs. Del/Ins + Del/Del	Location (cardia and non-cardia)	2.20 (1.26–3.84)	0.005
*miRNA4463_rs5877455_*	Ins/Ins vs. Del/Ins + Del/Del	Age (less or more than 40 years)	2.80 (1.64–4.80)	0.000

Genotypes Del/Del = homozygous deletion, Del/Ins = heterozygous, and Ins/Ins = homozygous insertion.

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
