# Peer review of "Association between INDELs in MicroRNAs and Susceptibility to Gastric Cancer in Amazonian Population"

_genes, 2022, doi:10.3390/genes14010060_

Round 1

Reviewer 1 Report

The authors investigated INDELs in eleven miRNA genes in gastric cancer patients and a healthy control group. The idea is interesting, the methods sound adequate and appropriate, and the manuscript is relatively well-written. However, some points remain to be addressed:

Comments:

-       Part 2.2. Please specify the type of specimen.

-       Part 2.3. Please specify the conditions for nucleic acid storage.

-  Part 2.4. Please provide a citation for “These miRNAs were found differentially expressed in gastric cancer in the literature”.

- I have concerns about the control group. What were their complaints/diseases that led them to be admitted to the hospital? How do the authors know that they were not affected by gastric cancer? Did the authors check the control group for any possible consanguinity/ family relationship with gastric cancer patients? These points may affect the results to some extent. If they are not addressed, please include them as limitations of the study in the discussion section.

-  The authors applied a single tube multiplex PCR with subsequent fragment size analysis with no fluorescence labels. The accuracy of most capillary electrophoresis instruments is +/- 3 bases and some amplicon sizes (regarding Table 1) are very close to each other (e.g. MIR920 and MIR3652). I would like to know the experience of the authors with this method. How reliable do they think this method is in defining alleles? Did they encounter any difficulties that made them perform an orthogonal method to confirm or rule out the presence of an allele?   

-  Table 1. The authors stated in lines 75-76 that they have chosen INDELs with a minimum allele frequency (MAF) ≥ 2%, but according to Table 1, the MAF for MIR302C is 0.08 which is less than 2%.

-  Table 1. I think the shorter allele for MIR302C and MIR920 are best curated as delins, rather than pure deletions.

-     Table 1. Please recheck the amplicon sizes for MIR920 and MIR3652. In addition, I think “pb” should be changed to “bp”.

-     Table 2. Why did the authors choose 40 years as the cutoff for age?

-   Some items in Table 2 and the Supplementary tables are not written in the English language. Please check them.

-    It is recommended to include tumor grade, T, N, M, and size of the tumor in Table 2 as well as in the statistical analysis with miRNAs alleles.

-   Table 3. Please specify what I and D stand for. It looks this is the first time they appear in the manuscript.

-  Table 3. I suggest separately reporting allele frequencies (I and D) for cases and controls and to provide their p-values.

-    The OR in Table 3 needs clarification. Is it for genotype frequencies or allele frequencies? What is its utilization here?

-  According to the supplementary Table S3, the numbers for MIRNA3652 DD should be written as 2,7/9,9.  

-  Table 4: I think the OR for MIRNA920 (0,43, 0.24-0.77) is not consistent with the corresponding genotype frequencies in Table 3, because ID+DD genotypes are more frequent in GC patients than controls. It might be a matter of reference category setting in logistic regression. Please clearly indicate the reference category (I/I?). The answer to this comment can potentially influence other parts of the manuscript.

-  Lines 240-241: Please recheck these two lines. “inhibiting silencing” of a tumor suppressor is not expected to lead to oncogenic activities.  

-  Table S5: I suggest also trying for T1/T2 versus T3 / T4. 

Reviewer 2 Report

I have only some minor comments:

-The Results should be detailed in some points (see 121-125;  140-145);

-Supplementary figures should be quoted in the text and should renumbered as S1,S2,S3;

-The paragraph "Dstribution of genotypes associated with clinical variants in GC" should be numbered as 3.3 and corrected as "Distribution of .......".

Author Response

-The Results should be detailed in some points (see 121-125; 140-145):

Response: Recommendations accepted and included.

-Supplementary figures should be quoted in the text and should renumbered as S1,S2,S3:

Response: Recommendations accepted and included.

-The paragraph "Dstribution of genotypes associated with clinical variants in GC" should be numbered as 3.3 and corrected as "Distribution of .......".

Response: Recommendations accepted and included.

Reviewer 3 Report

Line 39 has to be changed. It’s oct 2022 so 2020 estimate is obsolete.

Author Response

Line 39 has to be changed. It’s oct 2022 so 2020 estimate is obsolete.

Response: INCA – Instituto Nacional do Câncer, responsible for producing epidemiological data in Brazil, published the last estimate in 2020, which is valid for the biennium 2021 and 2022.

“... it was possible to produce information from this new volume of Incidence Estimates: Cancer Incidence in Brazil, for 2020-22, in which 19 specific cancer locations were considered” (INCA, 2022) - https://www.inca.gov.br/estimativa

Comments

Interesting paper that explore the susceptabity to CG and consequenty the possible role in early detection not excluding a possibible implication on the magament of Hereditary Sndrome. What about Helicobacter infection?

Response: Yes, we performed association tests with INDELs markers in MIRNAs, but we did not obtain relevant and significant results for H. pylori infection in this study (p<0.05).

Round 2

Reviewer 1 Report

I would like to thank the authors for replying to the comments and revising the manuscript. Some points remain to be addressed:

1. An asymptomatic person may still have early-stage gastric cancer. I wish the authors would have selected the controls from patients with a negative upper gastrointestinal endoscopy. Moreover, the authors did not check controls for possible consanguinity/ family relationships with the cases. These two points may affect the results slightly, but should not be overlooked. I suggest including them as limitations of the study in the discussion section.

2. I thank the authors for sharing their experience regarding the genotyping method (single tube multiplex PCR with fragment size analysis). However, I am afraid the use of fluorescent dyes is not mentioned in the present manuscript, while it is an illuminating point. Therefore, I suggest adding this fact (the use of fluorescence) to the manuscript.

3. Table 3: what I expect from an allele/genotype frequency analysis is to report the number (and percentage) of every allele (and genotype) frequencies in each group (case and control) with associated p-values and Odds Ratios (with their confidence intervals).   

Author Response

The lines cited in the responses to reviewer 2, in round 2, were misquoted, the correct citation would be:
Response 1: (lines 168-169)
Response 2: (lines 82-83)
